# Genome-Wide Comparison of the Target Genes of the Reactive Oxygen Species and Non-Reactive Oxygen Species Constituents of Cold Atmospheric Plasma in Cancer Cells

**DOI:** 10.3390/cancers12092640

**Published:** 2020-09-16

**Authors:** Hwee Won Ji, Heejoo Kim, Hyeon Woo Kim, Sung Hwan Yun, Jae Eun Park, Eun Ha Choi, Sun Jung Kim

**Affiliations:** 1Department of Life Science, Dongguk University-Seoul, Goyang 10326, Korea; hweewon96@dongguk.edu (H.W.J.); heejoo0923@dongguk.edu (H.K.); opopr5@dongguk.edu (H.W.K.); skskbby@dongguk.edu (S.H.Y.); 201511717@dongguk.edu (J.E.P.); 2Plasma Bioscience Research Center, Kwangwoon University, Seoul 01897, Korea; ehchoi@kw.ac.kr

**Keywords:** skin cancer cells, leukemia cancer cells, cold atmospheric plasma, KEGG pathway, reactive oxygen species

## Abstract

**Simple Summary:**

Cold atmospheric plasma is being applied to treat cancer by virtue of its preferential anti-proliferative effect on cancer cells over normal cells. This study aimed to systemically determine the distribution of target genes regulated by the reactive oxygen species and non-reactive oxygen species constituents of the plasma. After analyzing genome-wide expression data for a leukemia and a melanoma cancer cell line from a public database followed by experimental approaches, PTGER3 and HSPA6 genes were found regulated by the non-reactive oxygen species and non-reactive nitrogen species constituents of the plasma in the cancer cells. This study could contribute to elucidate the molecular mechanism how each physicochemical constituent of the plasma induces the specific molecular changes in cancer cells.

**Abstract:**

Cold atmospheric plasma (CAP) can induce cancer cell death. The majority of gene regulation studies have been biased towards reactive oxygen species (ROS) among the physicochemical components of CAP. The current study aimed to systemically determine the distribution of target genes regulated by the ROS and non-ROS constituents of CAP. Genome-wide expression data from a public database, which were obtained after treating U937 leukemia and SK-mel-147 melanoma cells with CAP or H_2_O_2_, were analyzed, and gene sets regulated by either or both of them were identified. The results showed 252 and 762 genes in H_2_O_2_-treated U937 and SK-mel-147 cells, respectively, and 112 and 843 genes in CAP-treated U937 and SK-mel-147 cells, respectively, with expression changes higher than two-fold. Notably, only four and two genes were regulated by H_2_O_2_ and CAP in common, respectively, indicating that non-ROS constituents were responsible for the regulation of the majority of CAP-regulated genes. Experiments using ROS and nitrogen oxide synthase (NOS) inhibitors demonstrated the ROS- and reactive nitrogen species (RNS)-independent regulation of PTGER3 and HSPA6 when U937 cancer cells were treated with CAP. Taken together, this study identified CAP-specific genes regulated by constituents other than ROS or RNS and could contribute to the annotation of the target genes of specific constituents in CAP.

## 1. Introduction

Cold atmospheric plasma (CAP), referring to plasma produced at low temperatures and atmospheric pressures [1], has attracted attention in both basic and clinical research fields for its preferential anti-proliferative effect on cancer cells compared with normal cells [2,3]. This phenomenon may be attributed to the physicochemical components of CAP, including reactive oxygen species (ROS), such as H_2_O_2_ and O_2_^−^, and reactive nitrogen species (RNS), such as NO and NO_3_^−^ [4,5]. In particular, targets of ROS including ROS-interacting proteins, signal transducers, target proteins, and cellular responses have been extensively studied in cancer research through single-gene [6,7] or genome-wide approach [8]. The representative ROS-regulated biomarkers of oxidative stress in cancer are as follows: TGF-β1, which induces tumor migration [9]; Nrf2, which stimulates Klf9, thus activating ERK1/2 [10]; VEGF, which increases angiogenesis [11]. RNS has also shown tumor cell-modulation activities in concentration-dependent ways. For example, NO mediated apoptosis of cancer cells by stabilizing p53 through phosphorylation of key serine/threonine moieties [12,13]. Key pathways such as DNA repair, PI3K/Akt signaling, ERK, TGF-β signaling, and HIF are regulated by NO in promoting cancer [14,15], many of which are also regulated by CAP [16,17,18].

There are also studies on various cancer cells treated with CAP. However, the target genes of CAP were identified just by examining a single or a few genes in most studies. Previous studies reported that SESN2 and HMOX1 were upregulated in melanoma and lymphoma cells by CAP, leading to the increase of apoptosis [19,20], whereas HSP90 and MMP-9 were downregulated in breast and colon cancer cells, leading to death or growth inhibition of tumor cells [21,22]. These deregulation processes were found to increase apoptosis or inhibit cell proliferation. Only a few studies have examined genome-wide expression changes after the treatment of cancer cells with CAP. In a study of U937 lymphoma cells, a network containing a number of heat shock proteins and anti-apoptotic genes were identified [20]. Another study revealed an association between CAP treatment of SK-mel-147 melanoma cells and the inhibition of migration and disorganization of the actin cytoskeleton [23].

CAP has been shown to induce genome-wide epigenetic changes including CpG methylation, microRNA, and histone modification [24,25,26]. Specifically, ESR1 and DNAJC8 were hyper- and hypo-methylated at their promoter by CAP in MCF-7 breast cancer cells, and this induced their upregulation and downregulation, respectively [27]. ESR1 encodes ERα which is a nuclear receptor triggering the expression of genes fundamental to tumorigenesis [28]. DNAJC8 is a member of heat shock proteins (HSPs), many of which are associated with cancer aggressiveness and prognosis [29]. The oncogenic miR-19a-3p was downregulated by CAP, resulting in the inhibition of the proliferation of MCF-7 cells [25]. CAP also modulated H3K4 (a histone modification associated with gene activation) occupancy by regulating the histone demethylase JARID1A of which dysregulation can lead to the altered gene expression and tumor progression [26,30].

Despite a number of studies showing the association of CAP with cancer cell growth, studies linking each CAP component to target genes or cellular responses are limited. Sato et al. compared the effect of H_2_O_2_ and CAP on the viability and gene expression profile of HeLa cells, having found H_2_O_2_ as one of the major factors responsible for inhibition of the cell viability [31]. Thus far, ROS or NOS inhibitors have been used to confirm whether the deregulation of gene expression by CAP is attributed to ROS or RNS. The expression of several genes such as PTEN [32] and TNFR1 [33] was found to be affected by the ROS inhibitor N-acetyl-L-cysteine (NAC) in CAP-treated cancer cells. On the other hand, the expression of PDGFRβ and TSP-1 [34] was affected by the NOS inhibitor N^ω^-nitro-L-arginine methyl ester hydrochloride (L-NAME). 

A limitation of the studies using inhibitors is that the inhibitors are restricted to only RONS among the various components of CAP. Furthermore, no genome-wide approach of using the inhibitors together with CAP is available. In this study, the genome-wide expression profiles of two cancer cell lines (U937 and SK-mel-147) were compared following H_2_O_2_ and CAP treatment. Microarray expression data were retrieved from a database, and a pool of genes significantly deregulated by H_2_O_2_ and CAP were extracted. The filtered genes were then divided into H_2_O_2_-specific and/or CAP-specific categories. The RONS- or CAP-specific regulation of selected genes was determined using RONS inhibitors. 

## 2. Results

### 2.1. CAP and H_2_O_2_ Target a Small Number of Common Genes

To identify the genes regulated by H_2_O_2_ and CAP, microarray expression data were retrieved from a public database (GEO datasets) [35] and significantly deregulated genes were analyzed. Data from the SK-mel-147 melanoma (GSE65972) and U937 leukemia cell lines (GSE10896) were adopted because these cell lines included data for both H_2_O_2_ and CAP. Moreover, all the cells showed reduced tumor cell motility, or presented increased expression of apoptosis-promoting genes at the given treatment conditions of CAP [20,23] and H_2_O_2_ [36]. Microarray data from other cell lines were excluded because they did not contain complete data, missing either CAP or H_2_O_2_ array data. After filtering statistically significant genes (*p*-value < 0.05 and |fold change| > 2), the gene sets were compared by the volcano plot (Figure 1A), Venn diagram (Figure 1B,C), and heatmap (Figure 1D) analysis. Overall, both H_2_O_2_ and CAP induced the deregulation of a considerable number of genes under all treatment conditions (214–12,483). Notably, in U937 cells, a higher number of genes were deregulated by H_2_O_2_ (1456–1550) compared with CAP (214–238). The Venn diagram showed 762 and 843 common genes in H_2_O_2_- and CAP-treated SK-mel-147 cells, respectively (Figure 1B). On the other hand, there were 252 and 112 common genes in H_2_O_2_- and CAP-treated U937 cells, respectively. Subsequently, the genes regulated by H_2_O_2_ and CAP were compared, which identified 143 target genes regulated by both H_2_O_2_ and CAP in SK-mel-147 cells; however, there were no common target genes in U937 cells (Figure 1C). Heatmap analysis showed that the target gene profiles of H_2_O_2_ and CAP were markedly different between the two cell types (Figure 1D).

### 2.2. MAPK Signaling and Cancer Pathways Are Significant for CAP not H_2_O_2_

We searched for genes that were specifically regulated by H_2_O_2_ or CAP. First, the deregulated genes were annotated based on the KEGG pathway analysis. As result, ‘Pathways in cancer’, ‘Metabolic pathway’, Glycolysis/Gluconeogenesis’, and ‘Carbon metabolism’ were annotated in both the H_2_O_2_- and CAP-treated groups, whereas ‘MAPK signaling pathways’ were annotated only in the CAP-treated group (Figure 2A). Next, genes that were common in both the U937 and SK-mel-147 cells treated with H_2_O_2_ and CAP were compared, which identified four and two genes, respectively (Figure 2B and Appendix A). Then, we focused on PTGER3 (prostaglandin E receptor 3) and HSPA6 [heat shock protein family A (Hsp70) member 6] as they were identified in CAP-treated cells but not H_2_O_2_-treated cells. PTGER3 and HSPA6 are known to inhibit tumor cell growth [37,38]. In the silico database analysis, (CCLE) [39] showed their low expression in various cancer types (Figure 2C). 

### 2.3. PTGER3 and HSPA6 Are Targets of CAP but not ROS or RNS

We investigated whether the regulation of PTGER3 and HSPA6 by CAP is attributed to the ROS or non-ROS constituents of CAP. This was accomplished by examining the expression of the genes in cultured U937 cells in the presence or absence of a ROS inhibitor. SK-mel-147 cells were not analyzed as they were not available from the public cell banks. Initially, the production of ROS was verified under various treatment conditions. H_2_O_2_ was treated at 100 μM, the same concentration as in the analyzed microarray experiment, which induced apoptosis of the U937 cells [36]. CAP was treated for 600 s at the condition described in our previous study, which also induced apoptosis of cancer cells [40]. As shown in Figure 3, U937 cells showed an increased level of ROS following H_2_O_2_ or CAP treatment, which was suppressed by the ROS inhibitor NAC.

The regulation of the six genes that were identified as H_2_O_2_- or CAP-specific regulated genes (Figure 2B and Appendix A) was further validated by RT-qPCR, except for MEG3, which was a long non-coding RNA. As shown in Figure 4A, PTGER3 and HSPA6 were upregulated in CAP-treated cells, which did not occur in H_2_O_2_-treated cells. Their upregulation was not significantly affected by either NAC or L-NAME (a NOS inhibitor), suggesting that the regulatory effect of CAP on both genes was not mediated by ROS or RNS. The deregulation of the three genes CTLA4, ABCC12, and REG4, which had been identified from the microarray data of H_2_O_2_-treated cells, was confirmed by RT-qPCR. A similar result was observed in CAP-treated cells, and their expression was restored by NAC (Figure 4B). L-NAME attenuated the effect of H_2_O_2_ and CAP on the expression of CTLA4 and ABCC12; however, there was no significant effect for REG4 expression. Ar gas alone did not induce a significant change of the gene expression compared to non-treatment (Appendix A). These results suggest that CTLA4 and ABCC12 may be regulated by ROS and RNS; however, REG4 may be regulated only by ROS. 

The CAP-specific but not H_2_O_2_-specific regulation of PTGER3 and HSPA6 was further examined by the Western blot analysis. As shown in Figure 5 and Appendix A, their expression was not significantly affected by H_2_O_2_ and NAC. However, the expression of the genes was increased by 119–169% by CAP, which was only partially suppressed by NAC, suggesting the presence of non-ROS constituents in CAP regulating PTGER3 and HSPA6. To determine whether PTGER3 and HSPA6 are involved in the inhibitory effect of CAP on cell proliferation, the cell growth of U937 cells was monitored after treatment with gene-specific siRNAs. The result showed increased cell growth following treatment with siRNAs for both PTGER3 and HSPA6 compared to the control siRNA, which was attenuated by CAP (Figure 6A, Appendix A, and Appendix A). Subsequently, the siRNA-treated cells were analyzed for apoptosis. Apoptosis was decreased by 31% and 26% in cells treated with siRNAs for PTGER3 and HSPA6 compared to cells treated with the control siRNA, respectively (Figure 6B,C). These results suggest that PTGER3 and HSPA6 stimulated U937 cancer cell growth, which may be inhibited by the non-ROS and non-RNS constituents of CAP.

## 3. Discussion

The use of CAP has been extended to the clinical cancer treatment as it can preferentially induce the cell death of cancer cells over normal cells [2,3]. In fact, previous studies with cancer cells ranging from the in vitro cell culture to preclinical trials have proven CAP to be promising for clinical application [41]. However, due to the diverse constituents of CAP, each of which provokes differential biological changes in different cell types, establishing standard CAP treatment strategies has been hindered. Therefore, elucidation of the underlying molecular mechanism of how each CAP constituent inhibits cancer cell growth is vital to establish cancer treatment strategies. This study was performed to associate the individual physicochemical components of CAP, especially ROS and non-ROS constituents, with target genes. Our initial analysis of microarray expression data revealed that both H_2_O_2_ and CAP deregulated the expression of a number of genes (252–843). 

A comparison of the heatmap profiles of U937 and SK-mel-147 cells revealed several patterns. Most notably, different dosages of H_2_O_2_ and CAP induced different responses for various genes especially in CAP-treated SK-mel-147 cells and H_2_O_2_-treated U937 cells, which showed opposite regulatory patterns. This differential effect under different CAP treatment conditions was also observed in previous studies. For example, MCF-7 and MDA-MB-231 cells differentially responded to CAP as demonstrated by the significantly different sets of deregulated genes with approximately 50% of the deregulated genes showing opposing patterns in terms of CpG methylation [24]. Recently, a study found that the CAP treatment for 600 s and 10 times of 60 s had opposing regulatory effects on the ZNRD1 gene [40]. Another study reported the induction of either the proliferation or apoptosis of cancer cells depending on CAP treatment conditions [42]. Notably, H_2_O_2_ and CAP shared no common target genes in U937 cells, whereas there were 143 common genes in SK-mel-147 cells. U937 cells presented a lower number of regulated genes by CAP, which may come from the unique cellular property of U937 growing in suspension, or from the CAP treatment condition. In addition, there may exist subtle differences in the experimental system including the CAP apparatus, chemical composition of the culture medium, and cellular physiology. Therefore, to better understand cell-specific responses, the standardization of each component is required. Notably, PTGER3 and HSPA6 were regulated by CAP but not H_2_O_2_ in U937 and SK-mel-147 cells. This observation, however, is just from a single treatment condition of CAP and H_2_O_2_. Therefore, an approach of elaborating treatment conditions, i.e., multiple H_2_O_2_ concentrations and different CAP treatment conditions, is needed to obtain the comprehensive regulatory pattern. In fact, H_2_O_2_ induces versatile cellular responses such as wound healing [43] and necrosis [44] depending on its concentration. Moreover, CAP generated from different conditions has either anti-proliferation or pro-proliferation activity on cancer cells [45]. 

PTGER3 is a G protein-coupled receptor for prostaglandin E2 (PGE2), which exerts its effects through downstream components of cell proliferation pathways such as MAPK/Erk [37]. The aberrant lower expression of PTGER3 has been associated with the biological hallmarks of several malignancies with negative clinical outcomes [46]. HSPA6, a member of the HSP70 family, is a ubiquitous molecule within cells that acts as a molecular chaperone under conditions such as stress, including carcinogenesis. The protein is overexpressed in various carcinoma types and may be correlated with aggressiveness and prognosis [47,48]. In contrast, extracellular HSP70 results in ROS production, which might initiate tumor cell necroptosis. HSPA6 was also reported to inhibit the proliferation and invasion of garlic extract-treated bladder cancer cells [38]. Although the expression of the two genes was not affected by either H_2_O_2_ alone or co-treatment with NAC, it was upregulated by CAP, which was attenuated by NAC. This suggests that ROS other than H_2_O_2_ in CAP may contribute at least in part to expression regulation. Previous studies indicated that multiple genes are regulated by either superoxide or nitric oxide. However, no study related to PTGER3 or HSPA6 is available. The use of other types of ROS or RNS and their inhibitors should help identify the responsible constituents. In the case of PTGER3, multiple sizes of isoforms are synthesized from a single gene [49]. Each RNA could be differentially regulated at the post-transcriptional level such as RNA stability or translational efficiency. This could be in part an explanation for the differential regulation of RNA and protein by CAP and NAC.

The three proteins, CTLA4, REG4, and ABCC12, which were regulated by both CAP and H_2_O_2_ have been known to be involved in tumor cell growth or drug resistance. CTLA4 (Cytotoxic T-lymphocyte associated protein 4) is an immunoglobulin superfamily cell adhesion molecule, exclusively expressed on lymphocytes [50]. REG4 (Regenerating family member 4) is a growth factor with mitogenic effect by stimulating MAPK and ERK1/2 [51]. Elevated levels of these two proteins are associated with poor overall survival in several cancers including breast carcinoma and gastric cancer. Studies related to the effect of CAP on the proteins have not been reported yet. The finding of downregulation of CTLA4 and REG4 by CAP and H_2_O_2_ can help establish the molecular mechanism of how CAP inhibits cancer cell growth. ABCC12 (ATP-binding cassette subfamily C member 12) is a membrane transporter of which expression is increased in cancer to confer multidrug resistance [52]. The regulation of ABCC12 by CAP gives us a hint that CAP may affect cancer cells by modulating membrane transporters involved in drug delivery.

A limitation of the current study is from the shortage of available genome-wide data in the public database. Data on ROS are abundant, covering various cancer cell lines including U937 (GSE10896), SK-mel-147 (GSE65972), and HepG2 (GSE47739). However, data on CAP and RNS are limited; hence, a complete dataset of CAP, ROS, and RNS could be assembled only for the two cell lines, U937 and SK-mel-147. It should be also mentioned that the verification of microarray expression data was performed only in the U937 cells, which obscures generalizing the concept that PTGER3 and HSPA6 were regulated by the non-ROS constituent of CAP. Confirmation of the results at gene as well as cell level in SK-mel-147 and other cell lines is needed. In addition, the signaling cascade from CAP to the target genes should be characterized to fully understand how a specific constituent in CAP causes cancer cell death. Specific KEGG pathways including PTGER3 and HSPA6 could provide insights into the molecular network. 

## 4. Materials and Methods

### 4.1. Data Mining and Bioinformatics Analysis

The expression profiling datasets were obtained from GEO DataSets (GSE10896, GSE76022, and GSE65972) of the NCBI [53]. The online software GEO2R [54]. was employed to divide the samples into two and more groups and to screen the differentially expressed genes (DEGs). The selection criteria were *p*-value < 0.05 and |expression fold change| ≥ 2. The online tool DAVID [55] and KEGG [56] were used for the functional annotation of DEGs. Clustering analysis was performed using the Clustering 3.0 software [57] with visualization using the TreeView program [58]. The expression profiles of PTGER3 and HSPA6 in various tumor types were based on the Cancer Cell Line Encyclopedia (CCLE) [39].

### 4.2. Cell Culture and Treatment with CAP

The human monocytic leukemia cell line U937 was purchased from the Korean Cell Line Bank (KCLB, Seoul, Korea) and cultured in RPMI-1640 medium (Welgene, Gyeongsan, Korea) supplemented with 10% fetal bovine serum (Capricorn Scientific, Ebsdorfergrund, Germany) and 2% penicillin/streptomycin (Capricorn Scientific) at 37 °C in a humidified incubator with 5% CO_2_. CAP was administered using a mesh-dielectric barrier discharge (DBD)-type device (Kwangwoon University, Seoul, Korea) as described previously [40]. In brief, cells in the culture medium were exposed to CAP (0.3 kV, 12.6 mA) for 600 s using argon gas flowing at 2.0 L/min with an operation frequency of 12.9 kHz. The cells were harvested 3 h after CAP treatment for further analysis.

### 4.3. Cell Transfection

To induce the downregulation of PTGER3 and HSPA6 in U937 cells, the cells were transiently transfected with siRNAs for those genes (Bioneer, Daejeon, Korea) or a negative control siRNA (siNC) (Bioneer) using serum-free Opti-MEM medium (Gibco BRL, Carlsbad, CA, USA) and Lipofectamine RNAiMAX (Invitrogen, Carlsbad, CA, USA) according to the manufacturer’s instruction. The cells were seeded at 1 × 10^4^ cells per well of a 96-well plate or 1 × 10^6^ cells per 60 mm dish with 100 µL and 2 mL medium, respectively, and transiently transfected with siRNA at a final concentration of 40 or 60 nM. The cells were then cultured for 24 h for RNA extraction and functional assays. The sequences of the siRNAs are shown in Appendix A.

### 4.4. ROS Detection

Generation of ROS and treatment of RONS inhibitors were achieved following a protocol previously described [59] with minor modifications. Briefly, the U937 cells were seeded at 1 × 10^6^ cells per 60 mm dish with a 2 mL medium. To induce ROS, H_2_O_2_ (Sigma-Aldrich, St. Louis, MO, USA) was added to the culture medium at a final concentration of 100 µM, and the cells were cultured for 18 h. NAC (Sigma-Aldrich), a ROS inhibitor and L-NAME (Sigma-Aldrich), a NOS inhibitor, was treated to cells in the culture medium at a final concentration of 2 mM for 3 h and 5 mM for 4 h, respectively, followed by H_2_O_2_ or CAP treatment. To detect intracellular ROS in U937 cells, the cells were incubated with 20 µM 2′,7′-dichlorofluorescin diacetate (DCFH-DA) (Sigma-Aldrich) in the RPMI-1640 medium for 1 h. The cells were then washed twice with PBS (Welgene) and finally suspended with 1 mL of RPMI-1640 medium. To detect ROS under a fluorescence microscope (Leica Microsystems, Wetzlar, Germany), 8 × 10^5^ cells were diluted in 1 mL of RPMI-1640 medium and transferred to a 60 mm culture dish. Three microscopic images were taken per sample to calculate the average fluorescence level. The fluorescence level was analyzed using ImageJ (National Institutes of Health, Bethesda, MD, USA). ROS was also detected using an Infinite 200 Pro fluorescence microplate reader (Tecan, Mannedorf, Switzerland) after diluting 5 × 10^3^ cells in 100 µL medium per well of a 96-well plate. The fluorescence was measured at an excitation wavelength of 485 nm and an emission wavelength of 530 nm.

### 4.5. Cell Proliferation Analysis

U937 cells (1 × 10^4^ cells) in 100 µL medium were seeded in a 96-well plate (SPL, Pocheon, Korea) and transiently transfected with siRNAs for PTGER3 and HSPA6 or scrambled siRNA at a final concentration of 40 or 60 nM. The cells were then exposed to CAP for 600 s and further cultured up to 24 h before cell proliferation analysis using Cell Counting Kit-8 (CCK-8) (Enzo Biochem, Farmingdale, NY, USA).

### 4.6. Apoptosis Assay

Apoptotic cells were quantified by flow cytometry using FITC Annexin V Apoptosis Detection Kit II (BD Technologies, Franklin Lakes, NJ, USA). The cells (1 × 10^6^ cells) were seeded in a 60 mm plate, transiently transfected with siRNA, and cultured for 24 h. Then, 1 × 10^5^ cells were stained with FITC Annexin V and Propidium Iodide (PI) (Sigma-Aldrich) according to the manufacturer’s instructions. The fluorescence was detected with a BD Accuri C6 flow cytometer (BD Technologies), and the data were analyzed with BD Accuri C6 software (BD Technologies).

### 4.7. Quantitative Reverse Transcription Polymerase Chain Reaction (RT-qPCR)

Total RNA was prepared from the cultured cells using the ZR-Duet DNA/RNA MiniPrep kit (Zymo Research, Irvine, CA, USA) and reverse-transcribed to cDNA using ReverTra Ace qPCR RT Master Mix (Toyobo, Osaka, Japan). The cDNA was amplified using KAPA SYBR FAST qPCR Master Mix (Kapa Biosystems, Wilmington, MA, USA) with an ABI 7300 instrument (Applied Biosystems, Foster City, CA, USA). The relative gene expression was calculated using the 2^–∆∆Ct^ method with glyceraldehyde-3-phosphate dehydrogenase (GAPDH) as an internal control. The PCR condition was as follows: Enzyme activation at 95 °C for 3 min; 40 cycles of denaturation at 95 °C for 3 s, annealing/extension at 60 °C for 40 s.

### 4.8. Western Blot Analysis

Harvested cells were washed with PBS and suspended in RIPA buffer (Thermo Fisher Scientific, Waltham, MA, USA) and 1% protease inhibitor cocktail (Thermo Fisher Scientific). The lysate was centrifuged at 4 °C for 10 min at 13,000× *g*, and the supernatant was recovered. Proteins (15 μg) were subjected to SDS-PAGE, blotted on a PVDF membrane (Sigma-Aldrich), and treated with antibodies overnight at 4 °C. The blot was then incubated with HRP-conjugated anti-rabbit IgG antibody (1:1000, GTX213110-01; GeneTex, Irvine, CA, USA) for 2 h. The bands were visualized using the West Save Gold reagent (Abfrontier, Seoul, Korea) and quantified using the Image Lab software (Bio-Rad, Hercules, CA, USA). The antibodies used were anti-HSPA6 (1:1000, NBP1-32761; Novus Biologicals, Centennial, CO, USA), anti-PTGER3 (1:1000, PAB25115; Abnova, Taipei, Taiwan), and anti-β-actin (1:1000, bs-0061R; Bioss, Woburn, MA, USA).

### 4.9. Statistical Analysis

All experiments were performed at least in triplicate, and the results are expressed as the mean ± SD. Statistical analysis was performed using the SPSS 23.0 software (SPSS, Chicago, IL, USA). Student’s *t* test was performed to analyze fluorescence level, RT-qPCR, Western blot, and apoptosis assay. *p*-value < 0.05 was considered statistically significant.

## 5. Conclusions

A genome-wide comparison analysis revealed that ROS and CAP targeted completely different sets of genes in U937 and SK-mel-147 cells with only a few genes in common. Moreover, each physicochemical component of CAP regulated a unique set of genes wherein the non-ROS constituents of CAP targeted HSPA6 and PTGER3, which showed anti-proliferative and pro-apoptotic activity in the cancer cells. This study is the first systemic study to annotate the target genes of the ROS and non-ROS constituents of CAP and could contribute to elucidating the regulatory mechanisms of CAP components.

## Figures and Tables

**Figure 1 cancers-12-02640-f001:**
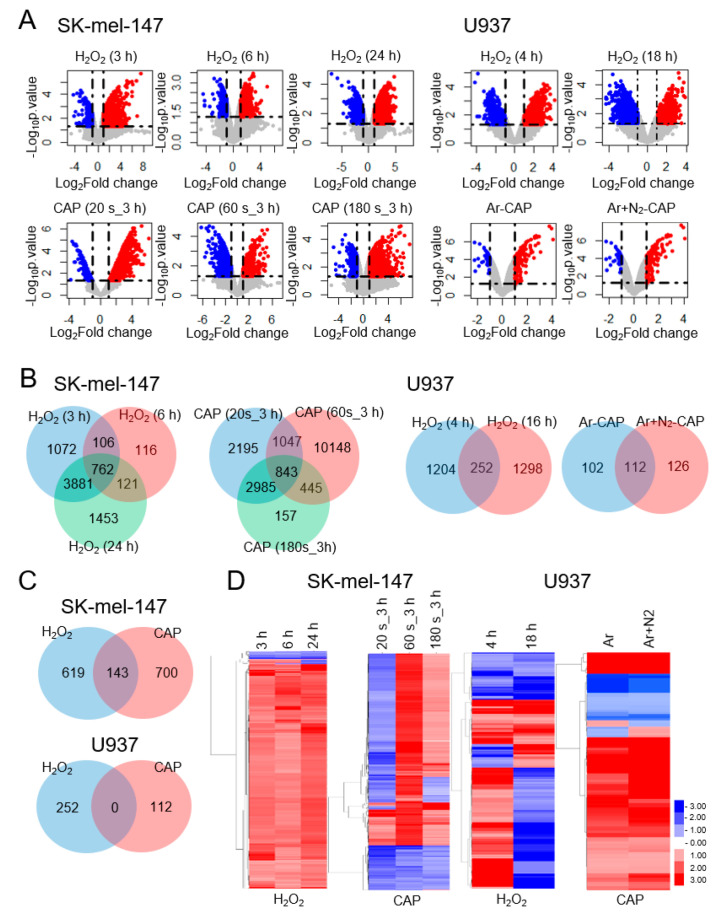
Genome-wide analysis of the target genes of H_2_O_2_ and cold atmospheric plasma (CAP) in cancer cells. The microarray expression data of U937 and SK-mel-147 cells treated with H_2_O_2_ or CAP were retrieved from GEO datasets. Genes showing significant expression changes (|fold change| > 2, *p*-value < 0.05) were filtered for further analysis. (**A**) Volcano plot of H_2_O_2_- and CAP-regulated genes. The indicated time is the treatment duration of cells with H_2_O_2_ (h) or CAP (s). Ar and N_2_ are flowed through gases while treating CAP. (**B**) Venn diagram of H_2_O_2_- and CAP-regulated genes. The number of genes significantly altered by H_2_O_2_ or CAP under different conditions (concentration or treatment duration) is represented by numerical values. (**C**) Venn diagram of the comparison of H_2_O_2_- or CAP-specific genes. Genes commonly appeared in the microarray data were classified as genes regulated by H_2_O_2_, CAP, or both. (**D**) Hierarchic analysis of H_2_O_2_- and CAP-regulated genes. Genes significantly altered by H_2_O_2_ or CAP are represented in a heatmap.

**Figure 2 cancers-12-02640-f002:**
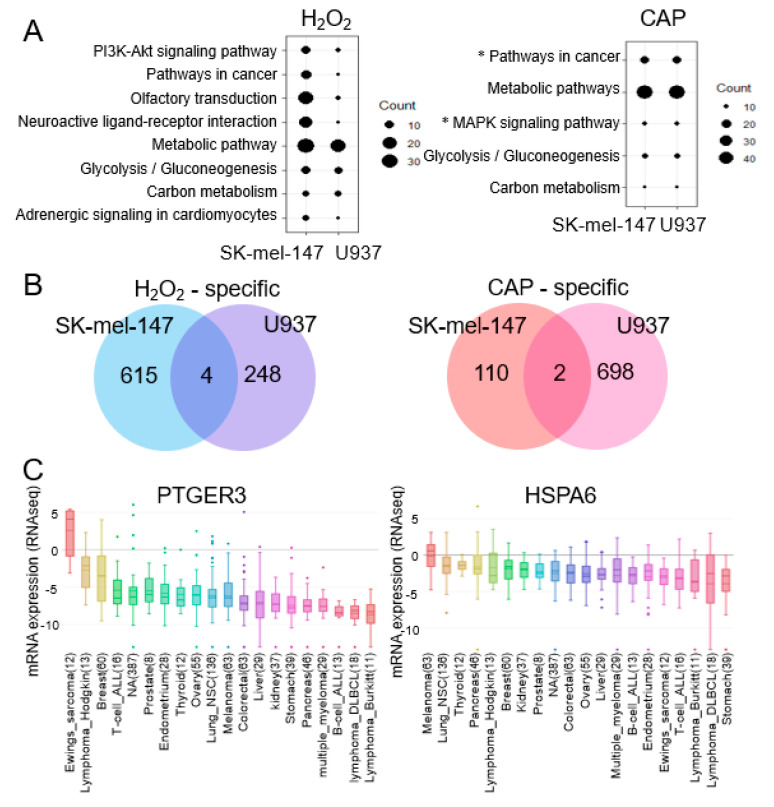
Pathway analysis of genes deregulated by H_2_O_2_ and CAP. (**A**) KEGG pathway analysis of genes regulated by H_2_O_2_ or CAP. Pathways observed in both U937 and SK-mel-147 cells are shown. The larger the size of the circle, the more genes included in the pathway. The ‘MAPK pathway’ and ‘Pathways in cancer’, which include PTGER3 and HSPA6 from CAP-treated cells, are indicated with an asterisk (*). (**B**) Venn diagram of common genes regulated in U937 and SK-mel-147 cells by H_2_O_2_ or CAP. There were four and two common genes significantly altered by H_2_O_2_ and CAP, respectively, in both cells. (**C**) Expression profile of PTGER3 and HSPA6 in various tumors. The expression level of the genes was extracted from the CCLE database and aligned with the box plot. The horizontal line indicates the expression level of control normal tissues. The numeric value in the parenthesis represents the number of examined tissues.

**Figure 3 cancers-12-02640-f003:**
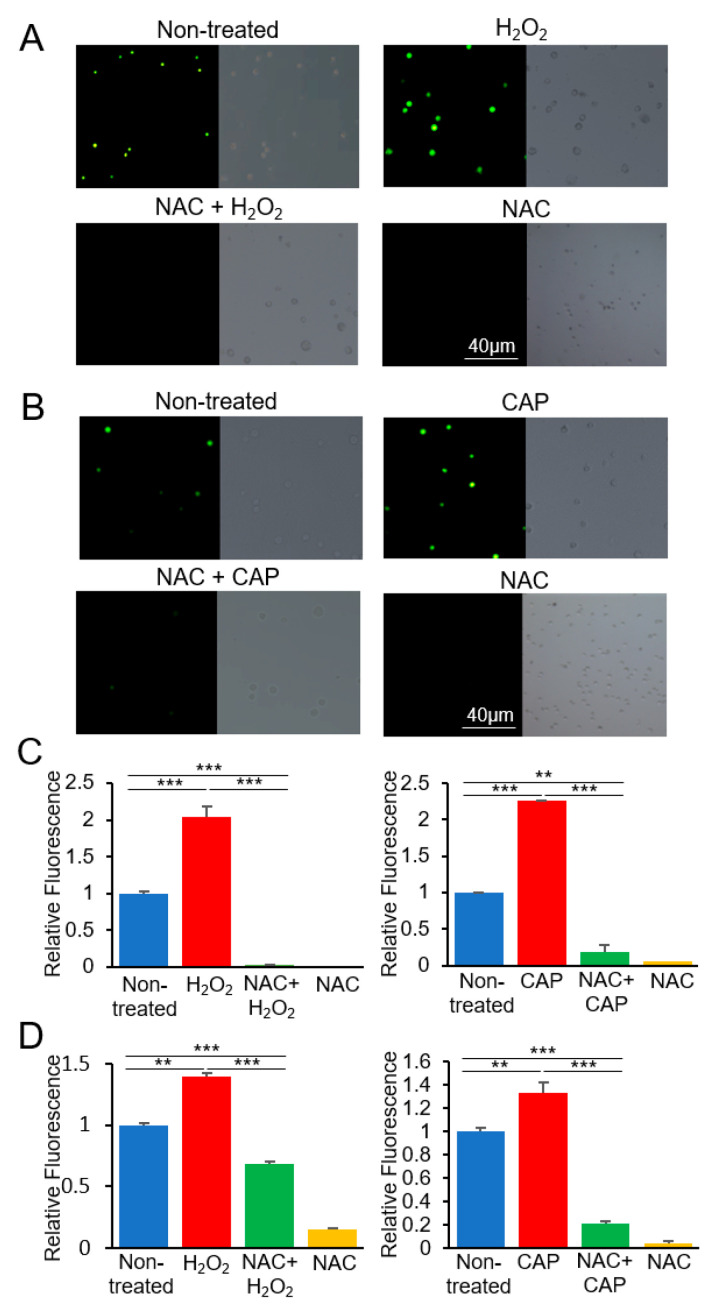
Production of reactive oxygen species (ROS) by H_2_O_2_ and CAP in U937 cells. U937 cells were treated with H_2_O_2_ or CAP with N-acetyl-L-cysteine (NAC), and the fluorescence intensity was observed using a microscope. Fluorescent images were obtained after treating the cells with (**A**) H_2_O_2_ and NAC or (**B**) CAP and NAC. Bright-field microscopy images are shown next to fluorescent images. Experiments were performed in triplicate, and representative images are shown. The bar graph shows the fluorescence intensity measured by ImageJ (**C**) and a microreader (**D**) (mean ± SE). Relative intensity of samples to the non-treated is indicated. ** *p* < 0.01, *** *p* < 0.001.

**Figure 4 cancers-12-02640-f004:**
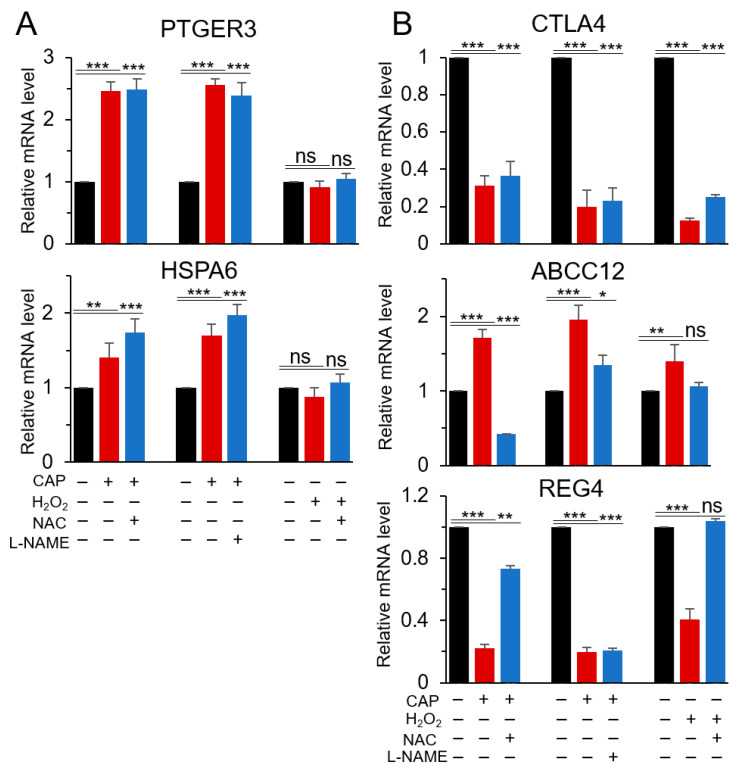
Effect of ROS or reactive nitrogen species (RNS) inhibitors on the expression of H_2_O_2_- or CAP-specific genes. To identify the physicochemical component responsible for the regulation of the five common genes regulated in U937 and SK-mel-147 cells by (**A**) CAP or (**B**) H_2_O_2_, their expression was examined by RT-qPCR using RNA from U937 cells cultured in the presence of a ROS inhibitor (NAC) or a NOS inhibitor (L-NAME). All assays were performed in triplicate, and the results are expressed as the mean ± SE. * *p* < 0.05; ** *p* < 0.01; *** *p* < 0.001; ns: Non-significant.

**Figure 5 cancers-12-02640-f005:**
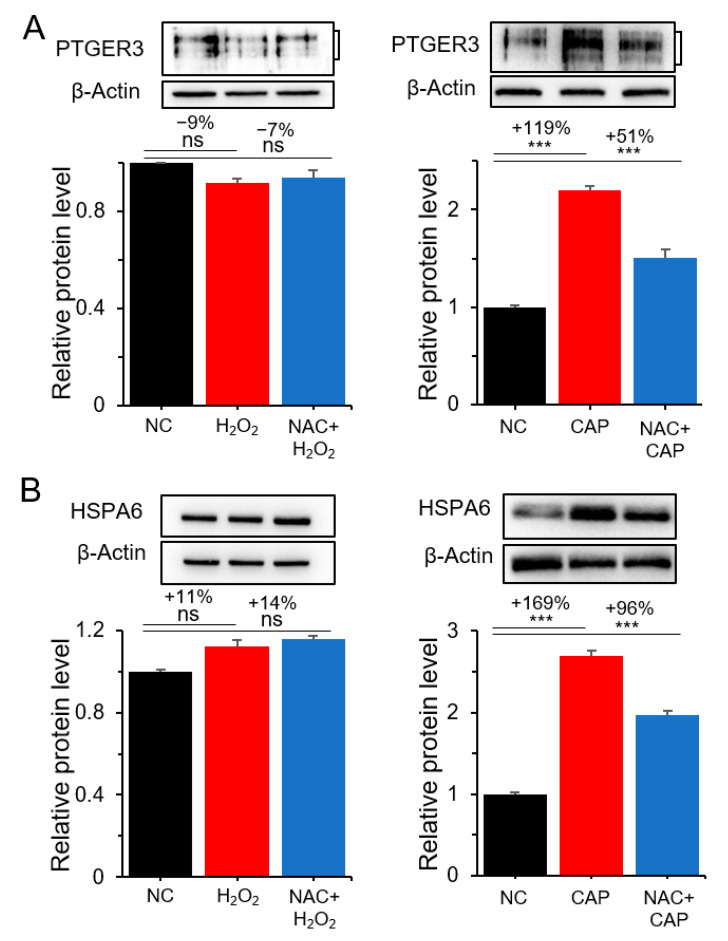
Regulation of PTGER3 and HSPA6 by the non-ROS components of CAP at the protein level. Western blot analysis was performed for PTGER3 and HSPA6 after treating U937 cells with H_2_O_2_ or CAP in the presence of NAC. (**A**) PTGER3 after H_2_O_2_ and CAP treatment with NAC. (**B**) HSPA6 after H_2_O_2_ and CAP treatment with NAC. All assays were performed in triplicate, and the results are presented as the mean ± SE. *** *p* < 0.001; ns: Non-significant.

**Figure 6 cancers-12-02640-f006:**
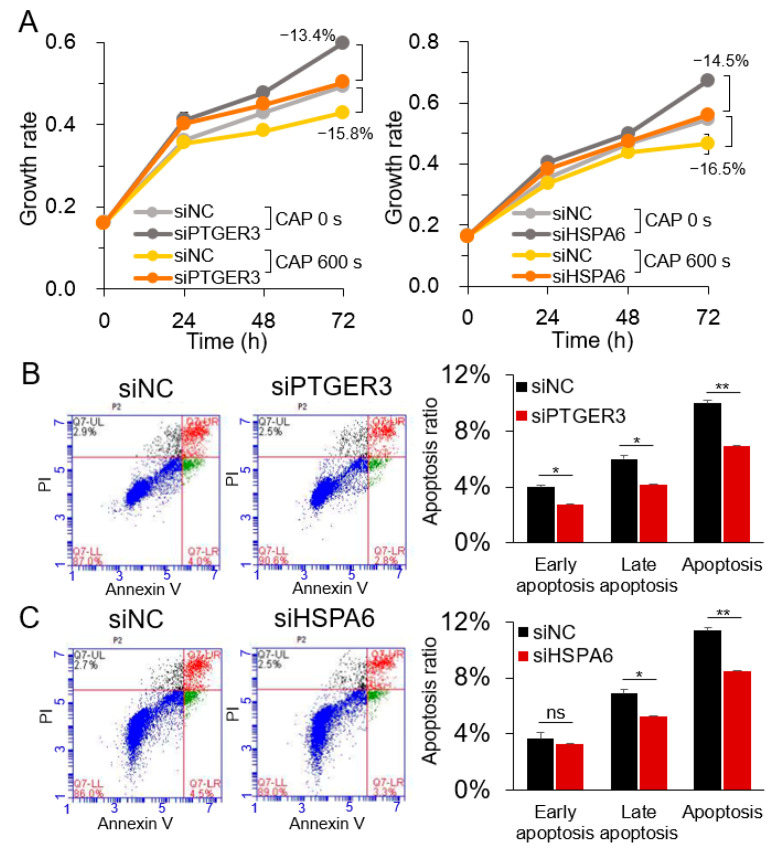
Suppression of U937 cancer cell growth by siRNAs for PTGER3 and HSPA6. PTGER3 and HSPA6 were downregulated in U937 cells by transient transfection with siRNAs, and the effect on cell proliferation and apoptosis was assessed by (**A**) CCK-8 assay and (**B**,**C**) flow cytometry analysis, respectively. siNC: Negative control siRNA; siPTGER3: siRNA for PTGER3; siHSPA6: siRNA for HSPA6. Representative images are shown for the flow cytometry analysis. All experiments were performed at least three times, and the results are presented as the mean ± SE. * *p* < 0.05; ** *p* < 0.01; ns: Non-significant.

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
