# Peer review of "Genome-Wide Comparison of the Target Genes of the Reactive Oxygen Species and Non-Reactive Oxygen Species Constituents of Cold Atmospheric Plasma in Cancer Cells"

_cancers, 2020, doi:10.3390/cancers12092640_

Round 1

Reviewer 1 Report

Accepted

Author Response

(Authors’ response) Thank you so much for your kind reviewing our manuscript. The authors appreciate your time.

Reviewer 2 Report

The authors have carefully addressed the reviewer’s comments, but I have still the following concerns.

The reviewer still concerns whether many comparisons, such as data analysis using microarray data from a public database and the authors' experiments, are reasonable. Besides, the comparison between CAP and H2O2 treatment is also doubtful. The authors selected two genes, PTGER and HSPA6, from the genome-wide comparison; indeed, CAP and H2O2 treatment showed different gene expression. However, both CAP and H2O2 treatments were performed with just one experimental condition. For example, does any concentration of H2O2 show different gene expression from that of any CAP treatment? The authors should describe the stringency of all comparisons more carefully. The reviewer thinks further experiments and discussions are required to lead the authors' conclusion.

The authors revised Materials and methods, but more accurate description is required. For example, several treatment conditions, such as cell numbers, treatment time, and incubation time, were performed. However, it is unclear which condition was applied to each analysis. The reviewer considers that the relationship between cell number and H2O2 concentration will affect the further analysis.

The authors revised Fig. 3. In Fig. 3C and D, the y-axis shows "Relative fluorescence." The criteria for relative values are not clear. Also, the number of observed cells should be described.

Fig. 6: The negative control is missing. The untreated control, which is not treated with CAP, should be performed. Besides, a similar experiment with H2O2 should also be conducted.

I hope that my comments will be very useful for the improvement of the article.

Author Response

The authors have carefully addressed the reviewer’s comments, but I have still the following concerns.

The reviewer still concerns whether many comparisons, such as data analysis using microarray data from a public database and the authors' experiments, are reasonable. Besides, the comparison between CAP and H2O2 treatment is also doubtful. The authors selected two genes, PTGER and HSPA6, from the genome-wide comparison; indeed, CAP and H2O2 treatment showed different gene expression. However, both CAP and H2O2 treatments were performed with just one experimental condition. For example, does any concentration of H2O2 show different gene expression from that of any CAP treatment? The authors should describe the stringency of all comparisons more carefully. The reviewer thinks further experiments and discussions are required to lead the authors' conclusion.

(Authors’ response) In this study, H2O2 of 100 uM was used to match the same concentration as in the analyzed microarray experiment, which induced apoptosis of the U937 cells. Treatment conditions of CAP used in this study was adopted from that in our previous study, because CAP also induced apoptosis of cancer cells at the condition. In the current manuscript, the authors focused on identifying genes regulated by the non-H2O2 constituent of CAP and revealed PTGER and HSPA6 to be regulated by CAP not by H2O2. However, as the reviewer pointed out, elaboration of experimental scheme is needed for precise comparison between H2O2 and CAP. Comparison at various H2O2 concentrations and CAP treatments followed by gene expression analysis would be helpful to compare the output between them. The authors added these facts in Result and speculated in Discussion with literatures that was kindly given by the reviewer in the comments to the initial submission.

The authors revised Materials and methods, but more accurate description is required. For example, several treatment conditions, such as cell numbers, treatment time, and incubation time, were performed. However, it is unclear which condition was applied to each analysis. The reviewer considers that the relationship between cell number and H2O2 concentration will affect the further analysis.

(Authors’ response) A missing information of experimental conditions for ROS detection was added in the Materials and methods where they should be specified. In addition, each treatment condition of different experiments was more detailed in the corresponding section to avoid confusion.

The authors revised Fig. 3. In Fig. 3C and D, the y-axis shows "Relative fluorescence." The criteria for relative values are not clear. Also, the number of observed cells should be described.

(Authors’ response) The fluorescence was normalized to the “Non-treated” sample. So, the values are relative ones to that of the “Non-treated” that was given a value of 1. This was described in the legend of figure 3. The number of cells for the experiment of figure 3 was indicated in the Materials and Methods (4.4 ROS detection). The measurement method of fluorescence from the microscopy was also specified.

Fig. 6: The negative control is missing. The untreated control, which is not treated with CAP, should be performed. Besides, a similar experiment with H2O2 should also be conducted.

(Authors’ response) In Fig. 6A, control experiments for CAP and the genes were performed. “CAP 0” stands for no treatment of CAP and “siNC” for a negative control of siRNA. The effect of CAP and H2O2 combined with siRNA on cell growth was performed only for CAP, because expression of the two genes, PTGER3 and HSPA6, was significantly affected by CAP but not by H2O2 as indicated in Fig.4 and 5.

I hope that my comments will be very useful for the improvement of the article.

(Authors’ response) The authors appreciate the precise and valuable comments of the reviewer.

Reviewer 3 Report

Authors partially addressed the concerns raised from the first revision of the manuscript. However, some points still need to be clarified before publication.

In the manuscript is still missing the reasoning behind the choice of H2O2 treatment condition (H2O2 100 uM, 18h) to be compared with CAP treatment. i.e. This H2O2 treatment condition is comparable to H2O2 produced after 10 min CAP treatment with the source analyzed by Authors? I think this is a crucial point to be discussed that will help the scientific community to weight up the results coming from this study, which contribute to the understanding that non-ROS species are involved in the anticancer activity of CAP, but cannot be exhaustive.

Introduction section was supplemented with additional literature information on the role of CAP, sometimes not easy to follow due to the specificity of regulated genes arising form other studies. However, the organization in gene differently regulated influencing cell viability, cell proliferation or epigenetic modification helps a little bit to focus on available information on the topic.

Lines 46-48: this sentence needs to be supplemented with literature on the effects of CAP on these molecular targets, which are partly reported later on, but a sentence should clarify that due to the role of these reactive species, CAP may play a role in counteracting cancer development.

Line 72: What Authors means with “inactivation of cell viability”? Do they mean inhibition? Please clarify.

Line 92: What Authors means with “retarded cell growth”? It is important to clarify, because it could be a temporary inhibition, and this point should be verified with recovery experiments that Authors did not performed. Please clarify.

Please specify in Figure 3 the treatment condition of NAC and if it is a pre-treatment or co-treatment, as I raised in the first revision of the manuscript. More, in material and methods some more details or references on the choice of the time of treatment with the inhibitor will help to support the experimental design proposed by Authors.

Please correct the proper name of NOS inhibitor: L-NAME, not L-LAME, as reported in some part of the manuscript.

Author Response

Authors partially addressed the concerns raised from the first revision of the manuscript. However, some points still need to be clarified before publication.

In the manuscript is still missing the reasoning behind the choice of H2O2 treatment condition (H2O2 100 uM, 18h) to be compared with CAP treatment. i.e. This H2O2 treatment condition is comparable to H2O2 produced after 10 min CAP treatment with the source analyzed by Authors? I think this is a crucial point to be discussed that will help the scientific community to weight up the results coming from this study, which contribute to the understanding that non-ROS species are involved in the anticancer activity of CAP, but cannot be exhaustive.

(Authors’ response) 100 uM of H2O2 was used following the same concentration as in the analyzed microarray experiment, which induced apoptosis of the U937 cells. Treatment conditions of CAP used in this study was adopted from that in our previous study, because CAP also induced apoptosis of cancer cells at the condition. However, as the reviewer pointed out, evidence of producing the same or similar level of H2O2 by CAP is missing. In fact, elaboration of experimental scheme by performing at various H2O2 concentrations and CAP treatments would be helpful to compare the output between them. These facts were supplemented in Result and the speculation was added in Discussion.

Introduction section was supplemented with additional literature information on the role of CAP, sometimes not easy to follow due to the specificity of regulated genes arising form other studies. However, the organization in gene differently regulated influencing cell viability, cell proliferation or epigenetic modification helps a little bit to focus on available information on the topic.

(Authors’ response) Thank you for the reviewer’s kind comments.

Lines 46-48: this sentence needs to be supplemented with literature on the effects of CAP on these molecular targets, which are partly reported later on, but a sentence should clarify that due to the role of these reactive species, CAP may play a role in counteracting cancer development.

(Authors’ response) As pointed out by the reviewer, the described pathways or genes are also regulated by CAP. Relevant literatures were given in the revision.

Line 72: What Authors means with “inactivation of cell viability”? Do they mean inhibition? Please clarify.

(Authors’ response) “Inhibition” is the correct expression. The authors appreciate correction of the reviewer.

Line 92: What Authors means with “retarded cell growth”? It is important to clarify, because it could be a temporary inhibition, and this point should be verified with recovery experiments that Authors did not performed. Please clarify.

(Authors’ response) “retarded cell growth” was corrected to “reduced tumor cell motility” based on the information in the literature.

Please specify in Figure 3 the treatment condition of NAC and if it is a pre-treatment or co-treatment, as I raised in the first revision of the manuscript. More, in material and methods some more details or references on the choice of the time of treatment with the inhibitor will help to support the experimental design proposed by Authors.

Please correct the proper name of NOS inhibitor: L-NAME, not L-LAME, as reported in some part of the manuscript.

(Authors’ response) NAC and L-NAME and was treated prior to CAP treatment. To avoid confusion, the treatment condition was specified in the Materials and methods (4.4. ROS detection). The misspelled L-LAME was corrected to L-NAME.

Round 2

Reviewer 2 Report

The authors have carefully addressed the reviewer’s comments. The reviewer recommends accepting this manuscript, but the reviewer has minor comments below for consideration.

The authors revised Fig. 3 and the Materials and Methods (4.4 ROS detection). The reviewer suggests that the number of observed cells (not seeded cells) to obtain Fig. 3C from Fig. 3A and B should be described.

Fig. 6: The negative control is missing. The untreated control, which is not treated with CAP, should be performed.
-> The reviewer apologizes for the confusing comment. There is no information about the experimental condition of Fig. 6B, and C.

The reviewer hopes that comments will be beneficial for the improvement of the article.

Author Response

The authors have carefully addressed the reviewer’s comments. The reviewer recommends accepting this manuscript, but the reviewer has minor comments below for consideration.

The authors revised Fig. 3 and the Materials and Methods (4.4 ROS detection). The reviewer suggests that the number of observed cells (not seeded cells) to obtain Fig. 3C from Fig. 3A and B should be described.

Fig. 6: The negative control is missing. The untreated control, which is not treated with CAP, should be performed.
--> The reviewer apologizes for the confusing comment. There is no information about the experimental condition of Fig. 6B, and C.

The reviewer hopes that comments will be beneficial for the improvement of the article.

(Authors’ response) The observed cell number was newly added in the Materials and Methods (4.4 ROS detection) of the revision.

As the reviewer suggested, an experimental design involving CAP would be helpful to understand the association of the genes and CAP in apoptosis. The authors performed Fig. 6B and C just to examine the effect of PTGER3 and HSPA6 on apoptosis after the genes had been proven to be regulated by CAP.

The authors appreciate the reviewer’s kind and accurate comments on their manuscript.

This manuscript is a resubmission of an earlier submission. The following is a list of the peer review reports and author responses from that submission.

Round 1

Reviewer 1 Report

The manuscript entitled "Genome-wide comparison of the target genes of the reactive oxygen species and non-reactive oxygen species constituents of cold atmospheric plasma in cancer cells" briefs about genome level changes in the presence of cold atmospheric plasma and H2O2 and delineates the role of ROS, RNS and non-ROS in the overall genetic changes observed. Evidence produced by the authors is substantial as it also hints the role of non-ROS in apoptosis and cancer cure. This will be a useful research work to understand the anticancer effects of cancer and I appreciate the authors for putting this together. I just have a couple of comments.

1) Please emphasize in the introduction why it is important it is to understand the role of non-ROS in anticancer effects caused by CAP. This is missing in the introduction part and may confuse the readers. This message was only given in the discussion part. Please revise.

2) What could be the reason for H2O2 dysregulating a higher number of genes than CAP itself (line 86)? Is it due to the exposure times (or) low concentration levels of H2O2 in CAP? Please comment.

3) I see a few punctuation and wording errors in the manuscript. Hence, I would suggest to proofread it carefully. Otherwise, the manuscript is well written. For ex. line 52, use "were" instead of "was".

Reviewer 2 Report

Manuscript 864134

Genome-wide comparison of the target genes of the reactive oxygen species and non-reactive oxygen species constituents of cold atmospheric plasma in cancer cells 

by HW Ji et al.

This work reports ROS-dependent and ROS-independent target genes of cold atmospheric plasma in the context of cancer treatment through genome-wide comparison.  It used a cold atmospheric plasma (CAP) in an argon flow at 2 liters per min and excited at 12.9 kHz for 10-min treatment of U937 leukemia and SK-mel-147 melanoma cells.  The two cells were also treated by 100-mM hydrogen peroxide (H2O2) for 18 h.  Common genes regulated by the CAP and H2O2 are considered as ROS-dependent genes and genes regulated by CAP but not H2O2 are regarded as ROS-independent target genes of the CAP.  Using NAC as an ROS scavenger and L-NAME as a RNS scavenger for U937 leukemia cells, PTGER3 and HSPA6 were identified as key target genes for CAP. 

ROS affect cell functions in a dose-dependent fashion, from regulation of signaling, through proliferation, to cell death.  For example, 20-min H2O2 at 0.5-50 mM signals for recruitment of neutrophils to wounds (Niethammer et al, Nature 459, 996 (2009)) and 1-h treatment of H2O2 at 16 mM is often used to induce necrosis (Dixon et al, Cell 149, 1060 (2012)). Gene expressions vary when cells are activated for different functions by H2O2.  Similarly, CAP affects cell functions, and associated regulation of gene expression, in a dose-dependent manner. It makes no sense to compare H2O2 at a dose for cell proliferation with CAP conditioned for cell death.

Unfortunately, the authors offer no explanation to their choice using 18-h, 100-mM H2O2 treatment to compare with 10-min CAP treatment. There is no diagnostic data of ROS and RNS in cell medium treated by the CAP. Therefore, there is no data with which to start to understand why 18-h, 100-mM H2O2 treatment wound represent the ROS in the CAP.  It is well known that argon plasma produces many other ROS, e.g. superoxide, singlet oxygen, hydroxyl radicals.  Given the above, it is difficult to see the value of the comparison.

Not withstanding the above, lets assume that some regulation of gene regulation by CAP are due to non-ROS ingredients. What are they?  The authors do not provide contributions by the argon flow, medium vaporization, heat, and electromagnetic wave.  If their contributions to the regulation of the gene expression are significant, then does this mean that one should use argon gas flow or medium vaporization as part of CAP-based cancer therapy? 

There are minor issues. For example, ref. 21 is cited as a CAP paper (line 56). This is incorrect.  The reader is referred to ref. 26 for details of the argon plasma used (line 246), but ref. 26 contains little info about the plasma.  Genome-wide regulation in CAP-treated cells has been done before, for example by the Greifswald group in the context of wound healing.  I believe that the authors overstate its importance.

Overall, this work has serious flaws in its study design and offers little value to the community. 

Reviewer 3 Report

Manuscript ID: cancers-864134

Submitted to: Cancers

Title: Genome-wide comparison of the target genes of the reactive oxygen species and non-reactive oxygen species constituents of cold atmospheric plasma in cancer cells

Authors: Hwee Won Ji, Heejoo Kim, Hyeon Woo Kim, Sung Hwan Yun, Jae Eun Park, Eun Ha Choi, Sun Jung Kim

This paper deals with identification of the target genes of specific constituents in cold atmospheric plasma (CAP). Medical applications of CAP, particularly cancer therapy, have recently increased due to its effective induction of apoptosis in a broad range of cancer cell types. The authors conducted genome-wide comparison of the target genes of the reactive oxygen species and non-reactive oxygen species constituents of CAP. However, On the other hand, I think that this paper still has some problems as indicated below. Therefore, this paper should be revised before further consideration.

  • p. 2, Introduction:

One of the earliest investigations of comprehensive gene expression analysis was performed by Sato, et al.. The following paper should be cited.

T. Sato, M. Yokoyama, and K. Johkura, "A key inactivation factor of HeLa cell viability by a plasma flow," Journal of Physics D: Applied Physics, vol. 44, no. 37, p. 372001, 2011. doi:10.1088/0022-3727/44/37/372001

  • My major concern is whether many comparisons, such as data analysis using microarray data from a public database and the experiments performed by the authors, are reasonable or not. The original experiments on the public database used different plasma source. Furthermore, the experiments performed by the authors are also different. H2O2 concentration is different too. In addition, the authors used 100 uM H2O2 as a comparison to CAP treatment. How did they determine the concentration? The authors should comment the above important points.
  • The quality of most of the figures should be improved. For example, x- and y- axis are unclear in Fig. 1 A, Fig. 2 C, Fig 6 B and C. The titles of y-axis are missing in Fig. 3 C, D, Fig. 4. All figures should be confirmed before resubmission.
  • More explanation on all results should be given. For example, what was happen in siNC cell after CAP treatment. All results should be mentioned in the manuscript.
  • p. 5: Relative protein level of PTGER3 is not agreed with mRNA level as shown in Fig. 4. Please explain about this. In addition, please show the protein levels of CTLA4, ABCC12, and REG4.
  • p. 9, L223: The authors mentioned about HepG2. Please show the result.
  • p. 9, L230-231: Dataset are shown, but original papers also should be cited.
  • Materials and methods should be described more precisely. For example, the timing of CAP treatment and the timing of addition of RONS inhibitors are unclear. Overall explanation should be reconsidered.

Minor comments

  • p. 1, Correspondence: Correspondence: [email protected]; Tel.: +82-31-961-5129
  • p. 1, Abstract, p. 2, L67-68: To my knowledge, L-NAME is a NOS inhibitor, not RNS.
  • p. 1, L39: NO3- -> NO3-
  • p. 3 caption: The counted number of the cells should be described.
  • Fig. 6 A: Horizontal axis Time -> Time [h]

I hope that my comments will be very useful for the improvement of the article.

Reviewer 4 Report

In the present manuscript Authors aimed to determine through a genome wide approach the genes differently regulated by ROS and non-ROS constituents of CAP in two cancer cell lines, U937 leukemia cells and SK-mel-147 melanoma cells. The analysis of gene expression data disclosed that CAP-regulated genes are mainly non-ROS constituents. The use of ROS and RNS inhibitors confirmed that the regulation of PTGER3 and HSPA6 was independent from the reactive species of CAP. Although the cutting-edge aim of the study, a revision is necessary before publication to help the readers focusing on the main findings of the study.

The manuscript aims to unravel the mechanism evoked by CAP to induce its anticancer effects, studying through a genome wide approach the genes differently regulated by CAP, by exploring the genes differently regulated by H2O2 and CAP treatment in the same cell lines. Although the potential of these new data, organization and clear explanation of emerging evidence is required, especially in the biological meaning of the identified genes.

In the introduction section it would be better to focus first on the role of reactive species in anticancer therapy and to list the main genes involved and identified via single genes or genome wide approach. Second, CAP should be introduced with special attention on the studies focusing on gene expression and plasma treatment. More, in this part it is not enough to provide a list of the genes differently regulated in cancer cell lines, but it would be nice to describe them at least placing them in the molecular and cellular pathway regulated by CAP, i.e. apoptosis and proliferation, metastatisation and angiogenesis, redox homeostasis. This organization would help to focus on the list of genes proposed by Authors and to clarify present background on this topic. For instance, the function of SESN2 or HMOX1 need to be clarified. In the same way, genes involved in the epigenetic modification regulated by CAP need to be better explained, i.e. ESR1 and DNAJC8; H3K4 and JARID1A. Some more details on the role of these genes will help to understand what is known about CAP-gene regulation, highlighting the key pathways.

Line 69: I think that limitation of the study should be better presented in discussion section.

Line 131: Please better explain why the expression of the emerging genes PTGER3 and HSPA6 were not analyzed in SK-mel-147. So why Authors chose this cell model for the comparison if not available? Please clarify.

In which treatment conditions were ROS detected? This is not described in the legend of Figure 3 nor in material and methods section. Please clarify.

Material and methods section:

Line 248: How Authors optimize the treatment conditions with H2O2? Please clarify. Usually H2O2 experiments are performed in PBS to avoid interaction with the components of medium and in particular with FBS that easily react with reactive species. To compare CAP and H2O2 the same experimental conditions should be kept, why 18 h after H2O2 exposure? An explanation on the choice of these treatment condition should be provided by the Authors.

Line 250: please better specify the treatment regimen with inhibitors. Are cancer cells pre-treated or co-treated respect to CAP exposure? Why NAC 3 h and L-NAME 4h? If cells are collected after 3h from CAP exposure, I think they are pre-treatment, but this is not specified. Please clarify.

Discussion section:

Line 185-187 need references and a deeper understanding on CAP selectivity. I agree on the statement by Authors that CAP mechanism of action need to be further addressed and that ROS and non-ROS species may give key contribution to this understanding, but this should be better expressed.

Results reported in Figure 4 need to be discussed. A critical revision of this results in the light of their hypothesized role in CAP mechanism of action should be presented by the Authors. These three genes are regulated both by H2O2 and CAP but critical consideration on their role is missing.

Lines 206 and line 209: to help the understanding of the results, the definition of PTGER3 and HSPA6 should be given earlier in the manuscript, already in the results section.

Line 213-214: Authors mentioned that HSP70 might initiate the non-canonical cell death, named necroptosis, but they do not explain this mechanism and they do not verify if the expression of HSPA6 can induce necroptosis, because only apoptosis was verified. Please clarify.  

It is well known that CAP anticancer effects depends on the synergistic activity of RONS and non-RONS constituents and this work may contribute to disclose the role of RONS-independent genes, but a deeper inside in the biological meaning of ROS-dependent and ROS-independent genes will help to better appreciate the results emerging from this genome wide study.

Please correct in the text: “an ROS inhibitor”